# Secretory and Membrane-Associated Biomarkers of Mast Cell Activation and Proliferation

**DOI:** 10.3390/ijms24087071

**Published:** 2023-04-11

**Authors:** Roberta Parente, Valentina Giudice, Chiara Cardamone, Bianca Serio, Carmine Selleri, Massimo Triggiani

**Affiliations:** 1Division of Allergy and Clinical Immunology, University of Salerno, 84081 Baronissi, Italy; roparente@unisa.it (R.P.); ccardamone@unisa.it (C.C.); 2Division of Hematology and Transplant Center, University of Salerno, 84081 Baronissi, Italy; vgiudice@unisa.it (V.G.); bianca.serio@sangiovannieruggi.it (B.S.); cselleri@unisa.it (C.S.)

**Keywords:** biomarkers, flow cytometry, immunophenotyping, mast cell activation, mast cells, mastocytosis, soluble mediators

## Abstract

Mast cells (MCs) are immune cells distributed in many organs and tissues and involved in the pathogenesis of allergic and inflammatory diseases as a major source of pro-inflammatory and vasoactive mediators. MC-related disorders are heterogeneous conditions characterized by the proliferation of MC within tissues and/or MC hyper-reactivity that leads to the uncontrolled release of mediators. MC disorders include mastocytosis, a clonal disease characterized by tissue MC proliferation, and MC activation syndromes that can be primary (clonal), secondary (related to allergic disorders), or idiopathic. Diagnosis of MC disorders is difficult because symptoms are transient, unpredictable, and unspecific, and because these conditions mimic many other diseases. Validation of markers of MC activation in vivo will be useful to allow faster diagnosis and better management of MC disorders. Tryptase, being the most specific MC product, is a widely used biomarker of proliferation and activation. Other mediators, such as histamine, cysteinyl leukotrienes, and prostaglandin D2, are unstable molecules and have limitations in their assays. Surface MC markers, detected by flow cytometry, are useful for the identification of neoplastic MC in mastocytosis but, so far, none of them has been validated as a biomarker of MC activation. Further studies are needed to identify useful biomarkers of MC activation in vivo.

## 1. Introduction

Mast cells (MCs), tissue-located immune cells of hematopoietic origin, reach their terminal differentiation and maturation in peripheral tissues, primarily in those that are in close contact with the external environment, such as the skin, gastrointestinal tract, and airways [1,2]. MCs are a heterogeneous cell population in terms of phenotype and profile of secreted mediators. The heterogeneity of the MC phenotype is acquired in peripheral tissue and is primarily due to exposure to specific tissue factors and microenvironmental stimuli. MCs are primarily activated by IgE bound to FcεRI, and by other stimuli, including complement products, viral and bacterial particles, and endogenous peptides [3]. In response to these and other stimuli, MCs can produce a wide array of mediators, including histamine, proteases, cytokines, and chemokines, involved in the initiation of innate immune responses and contributing to adaptive responses, such as defense against infections and parasites, suppression of tumor growth, and a balanced activation of these cells to maintain tissue homeostasis. However, dysfunctions in MC activation and functions potentially cause tissue damage and produce a cascade of events ultimately leading to several disorders, including allergic, autoimmune, and neoplastic diseases [4,5].

In the last decades, a group of clinical conditions primarily associated with MC dysfunction have been described, and are currently referred to as MC disorders characterized by increased MC proliferation and/or enhanced activation state leading to an excessive release of MC-derived mediators (Table 1). MC disorders can range from mild diseases with low-level or local (tissue-specific) mediator-induced symptoms, to severe and life-threatening conditions due to extensive neoplastic MC proliferation and/or massive and systemic mediator release. This variability in clinical phenotypes makes their initial diagnosis challenging, also because clinical pictures may mimic a variety of other diseases unrelated to MC. Therefore, the availability of reliable indicators of MC proliferation and activation is important to improve diagnosis and assess disease severity and responsiveness to treatment [6].

In physiological conditions, mature MCs do not circulate in peripheral blood, and plasma/serum and other biological fluids can be used only to measure circulating MC-released mediators. Bone marrow blood is a suitable specimen to study MC phenotype and functions by flow cytometry; however, the number of cells in patients with non-proliferative disorders is very limited. MCs can be also studied in peripheral tissues by immunohistochemistry; unfortunately, this technique has major limitations for the evaluation of the MC activation state.

In this review, we summarize the state-of-art on major MC-secreted mediators and on MC immunophenotypes, which can be used as potential biomarkers of cell proliferation and activation. There is a large consensus on the methodology to assess MC proliferation and on surface markers useful in differentiating between normal and neoplastic MCs [7]. Flow cytometry is a great tool to study surface MC markers to distinguish normal and neoplastic MC phenotypes. However, there is still a large area of uncertainty in the identification of surface molecules strictly associated with MC activation. The role of flow cytometry in the evaluation of the MC activation state is still unclear and represents an area of research.

## 2. Definition of MC-Related Disorders

MC-related disorders represent a wide spectrum of diseases characterized by abnormal MC proliferation and/or activation [8,9,10,11]. The list of MC-related disorders has increased over the last decades with better definitions of molecular abnormalities associated with these diseases. MC disorders currently include conditions in which cell proliferation is predominant, such as mastocytosis, and diseases in which cell activation is prevalent, such as mast cell activation syndromes (MCAS).

The mechanisms involved in MC activation are rather heterogenous and include IgE and non-IgE-dependent pathways. It is likely that in MC disorders, the two mechanisms are equally important. Activation of MC in mastocytosis and MCAS is probably the result of multiple stimuli involving several receptors expressed on the MC surface. These receptors include the classical high-affinity IgE receptor and other receptors activated by IgG, complement proteins, drugs, toxins, microbial components, and physical stimuli. It is important to note, however, that MC disorders may occur simultaneously in the same patient, e.g., a patient may have mastocytosis and concomitant MCAS or mastocytosis and hereditary alpha-tryptasemia. These overlap conditions are further influenced by the patient’s atopic state or other allergic comorbidities, which makes the clinical picture of MC diseases rather complex and heterogeneous.

### 2.1. Mastocytosis

Systemic mastocytosis (SM) is a rare disease characterized by clonal MC proliferation and infiltration in different organs. In a recent revision of the WHO classification, SM is considered an independent category of disease and no longer a subgroup of myeloproliferative neoplasms [11,12]. Disease severity and aggressiveness can vary from indolent to aggressive SM or MC leukemia [8]. Clinical heterogeneity is also related to a wide range of mediators that can be released by MC, such as histamine, tryptase, chymase, or other newly synthesized mediators [2,13]. Mastocytosis is caused by activating mutations in *KIT*, a gene encoding for Stem Cell Factor (SCF) receptor [14]. Various *KIT* mutations have been described, with the most frequent in adults being a point mutation involving a valine to aspartic acid substitution (D816V) in the catalytic domain, responsible for *KIT* constitutive activation leading to uncontrolled MC proliferation [14,15]. The current classification of mastocytosis distinguishes three clinical entities: cutaneous mastocytosis (CM) characterized by MC proliferation limited to the skin; systemic mastocytosis (SM) with extra-cutaneous involvement; and mast cell sarcoma (MCS) [12]. Clinical pictures are very heterogeneous and are characterized by combinations of symptoms caused by the release of mediators by MC and manifestations related to MC infiltration, leading to organ dysfunction. Organ infiltration and damage are predominant in advanced disease, whereas mediator-related symptoms could be present in all clinical entities [5]. Mediator-related symptoms are due to the release of several MC mediators, such as preformed molecules (histamine, heparin, tryptase, chymase) and newly synthesized substances (prostaglandins, leukotrienes, Platelet Activating Factor [PAF]) [13,16,17]. Indeed, MC can produce a wide range of molecules including cytokines such as tumor necrosis factor α (TNFα), interleukin-31 (IL-31), interleukin-6 (IL-6), and other cytokines with pro-inflammatory and immunomodulatory properties contributing to the great variability of clinical presentation [18,19,20].

### 2.2. Mast Cell Activation Syndromes

MC activation syndromes (MCAS), a heterogeneous group of disorders with various etiologies, present with episodic symptoms caused by the acute or chronic release of MC mediators [9,21,22]. Diagnostic MCAS criteria have been recently revised and include (i) evidence of acute and recurrent symptoms of MC activation, occurring simultaneously in at least two organs; (ii) increased MC mediator levels, especially tryptase; (iii) and clinical resolution or substantial symptom improvement after anti-mediator treatment [23,24,25]. All three criteria must be fulfilled for MCAS diagnosis.

The classification and main features of MCAS are shown in Table 1. Anaphylaxis is the most severe clinical expression of MCAS and is defined as a potentially life-threatening reaction characterized by acute onset of compromised breathing and/or circulation; it may also occur in the absence of typical skin manifestations (e.g., pruritus, urticaria, flushing, and angioedema) [26]. The role of MC in the pathogenesis of anaphylaxis is clearly established, and a variety of MC-derived mediators are involved in its clinical features.

### 2.3. Hereditary Alpha-Tryptasemia

A unique recently described form of MC-related disorder is the hereditary alpha-tryptasemia (HAT), a new genetic trait characterized by an increased number of a gene encoding for α-tryptase (*TPSAB1*), with an estimated frequency of 4.4–6% in the general population [27]. HAT patients have elevated basal serum tryptase and can present with a wide range of clinical manifestations, including Hymenoptera venom-triggered severe anaphylaxis, connective tissue abnormalities, autonomic dysfunction with orthostatic hypotension, and postural orthostatic tachycardia syndrome [27]. Interestingly, HAT prevalence is higher in SM patients, and this genetic trait is found in 17.2% of patients [28], which was also confirmed in an Italian multicentric study [29]. HAT represents a major risk factor for severe anaphylaxis in SM patients as SM patients with HAT show more severe mediator-related symptoms compared to SM without *TPSAB1* alterations [28]. Therefore, the evaluation of *TPSAB1* mutational status can identify patients at higher risk of anaphylaxis [30].

Table 2 shows the main MC disorders and illustrates their characteristics in terms of MC burden, associated mutations, expression of clonal markers, and tryptase levels.

## 3. Soluble Biomarkers of MC Activation

A mandatory criterion for diagnosis of MCAS is the assessment of validated biomarkers of MC activation [6,12,31,32]. MC proliferation and activation markers (both validated and under evaluation) can be divided into two subgroups: secreted mediators, measurable in biological fluids; and cell surface markers expressed by MC isolated from different tissues and usually examined by flow cytometry. Figure 1 illustrates the circulating molecules and their metabolites as well as the surface markers used to assess MC activation and proliferation.

### 3.1. Preformed Mediators

#### 3.1.1. Histamine

Histamine, produced by a histidine decarboxylase that removes a carboxylic acid residue from L-histidine, is the major preformed MC mediator and it is stored in large quantities within secretory granules. Histamine predominantly binds to H_1_ and H_2_ receptors expressed on endothelial bronchial smooth muscle and gastric parietal cells, respectively, and causes skin wheal and flare reactions, bronchoconstriction, hypotension, vasodilation, increased capillary permeability, and gastrointestinal hyperactivity [31,32,33,34,35]. Histamine is also produced by basophils and other cell types, and, once released, is rapidly metabolized by several enzymes, such as histamine N-methyl transferase and diamino-oxidase.

Histamine is considered the central mediator involved in the pathogenesis of anaphylaxis and in the past has been proposed as a marker of anaphylactic events. Indeed, plasma levels of histamine could be elevated in patients with anaphylaxis; however, it should be optimally measured within 15–60 min after the onset of symptoms [35,36]. During anaphylaxis, the entity of histamine elevation is higher in patients with cutaneous symptoms (erythema, urticaria, and angioedema) and respiratory findings (wheezing) [36]. Due to the short half-life and circadian variations, measurement of plasma histamine is difficult and not routinely employed in clinical practice [6,36,37]. For these reasons, measurement of histamine metabolites in 24 h urine samples is recommended [31]. The most common metabolites measured in clinical practice are N-methyl histamine (N-MH), a histamine N-methyltransferase product, and 1-methyl-4-imidazole acetic acid (MIMA), a diamino-oxidase metabolite [38]. Urinary N-MH is currently measured by liquid chromatography/tandem mass spectroscopy and is expressed as mcg/g of creatinine (Cr). Normal values depend on age and in adult patients vary from 30 to 200 mcg/g Cr. However, urinary histamine metabolite determination has some pre-analytic limitations. Measurement of these metabolites could be influenced by the intake of foods containing biological amines, such as bananas, cheese, spices, or spinach, and by bacterial contamination or storage conditions of urine samples [31]. Moreover, the lack of standardized normal ranges is another drawback to the routine use of urinary histamine metabolites in clinical practice. Urinary N-MH is potentially useful in the diagnosis of patients with suspected mastocytosis, especially in patients without typical skin involvement and with only a slight elevation of serum tryptase (10–20 μg/L). In these subjects, diagnosis may be difficult, and the evidence of high N-MH levels may lead to performing a bone marrow (BM) biopsy [39]. In patients with mastocytosis, histamine metabolites correlate with MC infiltration in the BM and with serum tryptase [31,40]. Therefore, histamine metabolites should be considered as a non-invasive marker of MC proliferation, while there is less evidence of their role as markers of symptom severity in mastocytosis. Urinary N-MH levels greater than 400 mcg/g Cr correlate with BM infiltration and mutated *KIT* Asp816Val allele burden in MC-related disorders [41].

Urinary histamine metabolites are proposed as a diagnostic tool in patients with suspected MCAS [6], together with other mediators, such as urinary prostaglandins and leukotrienes E4. A major limitation of histamine metabolite assays in MCAS is the lack of validated thresholds to define a significant MC activation in vivo.

Table 3 summarizes the main advantages and potential limitations of measuring histamine metabolites and other MC-derived soluble mediators or their products.

#### 3.1.2. Tryptase

Tryptase, a serine protease, is the most specific biomarker of mastocytosis because almost all the circulating enzyme is derived from MC granules, and only a negligible amount is released by basophils [42,43]. Tryptase exists in two different isoforms: α-tryptase and β-tryptase. The first is secreted at a low rate by piecemeal MC degranulation, whereas the β-isoform is released after MC activation as mature tetramers of β-tryptase. Once MCs are activated and tryptase is secreted, serum peak is reached after 30–90 min with a half-life of about 2 h. Therefore, circulating tryptase levels should be evaluated within 4 h from symptoms to assess MC activation in MCAS or anaphylaxis [44]. The biological effects of tryptase isoforms are not yet completely defined. Tryptase induces vascular permeability, neoangiogenesis, inflammation, airway hyper-reactivity, fibrosis, and fibrinolysis, and has anticoagulant activities [44]. Moreover, tryptase induces the generation of bradykinin from kininogen that, in turn, increases vascular permeability and platelet activating factor (PAF) production by endothelial cells. Tryptase also promotes leukocyte recruitment by stimulating the protease-activated receptor (PAR2) family on endothelial cells [45,46]. However, most of this evidence is derived from in vitro studies, thus the real contribution to symptom development in MCAS remains still undefined and there are no clinical studies that conclusively allow a direct association between the effects of tryptase and symptoms. In HAT, α/β tryptase tetramers can directly activate PAR2 and EMR2 (EGF-like module containing mucin-like hormone receptor 2) receptors on endothelial cells, resulting in enhanced vascular permeability [47].

Median basal serum tryptase in healthy individuals is about 5 μg/L [48]. Several studies demonstrate an elevation of serum tryptase during anaphylaxis: in this case, measurement of tryptase should be performed within 1–4 h after clinical symptom onset.

During anaphylactic events, higher levels of serum tryptase were seen in patients with urticaria and tachycardia [36]. Moreover, there is a positive correlation between the severity of anaphylaxis and the degree of increase in serum tryptase [49]. The entity of elevation can vary based on the types of antigens causing anaphylactic reactions: serum tryptase may be only slightly elevated in patients with food-induced anaphylaxis in contrast to Hymenoptera venom-related anaphylaxis [50]. Despite this evidence, tryptase is not an ideal marker of anaphylaxis, because it remains low in 30–40% of patients with anaphylaxis [50]. The measurement of mature (tetrameric) β-tryptase could be used to determine the amount of tryptase released during MC degranulation.

In clinical practice, tryptase is the main tool for MC proliferation assessment. Tryptase determination is important to diagnose SM, as basal serum tryptase ≥20 μg/L is a minor criterion for SM diagnosis according to the WHO criteria [12]. Tryptase is higher in SM patients compared to those with cutaneous forms, and tryptase levels correlate with disease severity [12,51,52]. Tryptase has also a role in the staging of patients with mastocytosis: a serum tryptase ≥200 μg/L is considered a “B-finding” and suggests a high MC burden in these patients [12]. Two B-findings may allow the diagnosis of smoldering systemic mastocytosis (SSM), a specific disease variant. In addition, tryptase has a prognostic value, as a serum tryptase ≥125 μg/L is a negative prognostic factor for overall survival in patients with advanced SM according to the International Prognostic Scoring System for SM (IPSM) [53]. Finally, tryptase is considered one of the most reliable indicators of responsiveness to cytoreductive treatments, and total tryptase levels can be easily monitored in patients with advanced SM under cytoreductive therapies [54,55].

Correlations between serum tryptase levels and symptom severity in patients with mastocytosis are less clear and still under investigation. For example, higher basal tryptase levels are not directly correlated with the increased prevalence of Hymenoptera venom anaphylaxis in SM patients [56].

Tryptase levels are also found elevated in the acute phase of anaphylaxis, in chronic urticaria, end-stage renal failure, and in several hematologic diseases, such as acute myeloid leukemia, myelodysplastic syndromes, or *FLP1L1*-*PDGFRA*-mutated hyper-eosinophilic syndrome, as myeloid blasts are a source of tryptase [31,32,57].

HAT is an autosomal dominant genetic disorder characterized by the presence of two or more copies of the gene encoding for α-tryptase (*TPSAB1*) and serum baseline tryptase levels greater than 8 μg/L [27]. Due to the higher prevalence of this genetic trait among SM patients in comparison with the general population, evaluation of *TPSAB1* mutational status could be useful to avoid overestimation of MC burden in mastocytosis patients [28].

Tryptase is considered the most reliable and specific marker of MCAS in clinical practice. The EU/US consensus group on MCAS considers the evaluation of serum tryptase during acute events as an important tool for the diagnosis of MCAS. In particular, the panel group indicates an elevation of serum tryptase by at least 20% over individual baseline plus 2 μg/L absolute (measured within 2–4 h for acute symptoms) as diagnostic criteria of MCAS [23,25].

Since patients with SM and HAT can present with variability in basal serum tryptase over time, a panel of experts recently suggested a way to optimize the recognition of anaphylaxis in these patients. According to these authors, the use of an acute tryptase/basal tryptase ratio of 1.685 improves the specificity of the 20 + 2 rule to help clinicians diagnose anaphylaxis in patients with HAT and/or SM [58].

#### 3.1.3. Chymase

Chymase, a neutral serine protease stored in secretory MC granules, has various biological effects, including activation of angiotensin I, stimulation of mucus production, leukocyte recruitment, and enhancement of histamine functions after MC activation. Serum chymase increases during anaphylaxis up to 24 h after the start of reactions and correlates with tryptase [59]. A chymase increase in patients with anaphylaxis has been confirmed in studies conducted on autopsy cases of fatal anaphylaxis [60].

Only limited data are available on chymase, assessed as angiotensin I cleavage activity, measured in the serum of subjects with mastocytosis. In SM, chymase activity could be detected, with the highest activity in those patients with advanced diseases suggesting a relationship between chymase activity and MC burden, albeit chymase activity does not correlate with tryptase levels [61].

### 3.2. De Novo Synthesized Mediators

#### 3.2.1. Prostaglandin D_2_

Prostaglandin D_2_ (PGD_2_) is synthesized from prostaglandin H_2_ by PGD_2_ synthase after MC activation and contributes to increased vascular permeability, leukocyte recruitment, and neuronal activation [31,32,62]. However, PGD_2_ is not a specific MC biomarker because eosinophils and other inflammatory cells can also produce this mediator.

The main metabolites of prostaglandins are 11-β-prostaglandin F2α (11β-PGF2α) and 2-3-dinor-11β-PGF2α (BPG), both usually measured in 24 h urine specimens or in extemporary samples by liquid chromatography–tandem mass spectrometry.

Levels of PGD_2_ metabolites are elevated in the sera of patients with anaphylaxis: these mediators exhibit relatively high stability in body fluids during anaphylaxis and up to two hours after an anaphylactic episode [63]. Moreover, PGD_2_ metabolite levels correlate with the severity of anaphylaxis [63].

Prostaglandin metabolites are also elevated in patients with MC-related disorders. For example, 11-β-prostaglandin F2α (11β-PGF2α) is increased in most patients with MCAS and correlates with symptoms, especially with flushing [62]. Therefore, patients with mastocytosis or MCAS and elevated 11β-PGF2α can benefit from treatment with aspirin because of its irreversible inhibition of both cyclooxygenase (COX) 1 and 2 activities preventing the conversion of arachidonic acid to prostaglandins. Indeed, aspirin treatment might decrease or normalize 11β-PGF2α levels in patients with mastocytosis and mediator-related symptoms [64,65]. In the study by Ravi A et al., 11β-PGF2α was found elevated in 82% of patients with MCAS, particularly in those with flushing and pruritus [65].

Urinary excretion of 11β PGF2α above 3500 ng/24 h (reference range <1000 ng/24 h) in patients with SM corresponds to a high BM infiltration and correlates with the presence of *KIT* mutation. However, normal ranges are not well established, and 11β-PGF2α levels are reliable only in association with other biomarkers of MC activation, such as urinary histamine metabolites [31,32].

#### 3.2.2. Platelet Activating Factor (PAF)

PAF is a lipid mediator produced by activated MC, endothelial cells, leukocytes, and platelets. PAF causes a rapid increase in vascular permeability and smooth muscle contraction of airways and coronary arteries and contributes to hemodynamic alterations during anaphylaxis [66,67]. PAF synthesis seems to be essential in the development of anaphylaxis, and recombinant PAF acetyl-hydrolase may protect from anaphylaxis episodes in animal models [49]. Measurement of PAF in biological fluids is extremely difficult due to its rapid catabolism by plasma acetyl hydrolase (AH). Assays of PAF and its metabolite lyso-PAF can be conducted by liquid chromatography/mass spectrometry, not routinely employed in clinical practice despite its potential utility.

Serum PAF concentration measured during anaphylaxis correlates with reaction severity, and low activity of PAF-AH is associated with a severe course of anaphylactic reactions [49]. In the study by Vadas et al., serum PAF levels were higher in patients with hypotension compared to patients with symptoms limited to the skin (e.g., angioedema). This finding may be attributable to the PAF-dependent production of nitric oxide [49]. Moreover, serum PAF correlates with the severity of anaphylaxis better than histamine [68]. While PAF release was shown in blister fluid from a patient with diffuse cutaneous mastocytosis [69], there are no data on the release of this mediator in SM patients.

#### 3.2.3. Cysteinyl Leukotrienes

The family of cysteinyl leukotrienes (CysLT), including leukotriene C4, leukotriene D4, and leukotriene E4, promotes neutrophil and eosinophil rolling and recruitment and regulates vascular permeability and smooth muscle contraction [31]. CysLTs are released by MC, basophils, myeloid dendritic cells, eosinophils, and macrophages. After MC activation, leukotriene C4 is converted into leukotriene E4, which is secreted and contributes to symptom development in SM patients.

Urinary leukotriene (LTE4) is measured by mass spectrometry and expressed as picograms of LTE4 per milligram of creatinine (reference range among normal volunteers <104 pg/mg Cr). Increased urinary LTE4 levels have been reported during anaphylactic reactions and aspirin-exacerbated respiratory disease. The maximum increase in LTE4 occurs 3–6 h after an acute episode and returns to normal within 20 h. Urinary LTE4 is higher in patients with more severe anaphylaxis with marked hypotension compared to those with anaphylaxis without cardiovascular symptoms [70]. An elevation of urinary LTE4 was seen also in exercise-induced anaphylaxis [62]. Urinary LTE4 can be also elevated in SM compared to healthy controls and correlates with serum tryptase and histamine metabolite levels, while not with 11β-PGF2α [71].

#### 3.2.4. Cytokines

##### IL-6

IL-6, a cytokine with pleiotropic effects on different tissues/organs, plays a role in hematopoiesis, inflammation, bone metabolism, and immune responses [72]. IL-6 is produced by MC, and patients with mastocytosis have increased IL-6 plasma levels probably related to the constitutional activation of the *KIT* signaling pathway caused by D816V mutation, as described in in vitro models [72,73]. Plasma IL-6 levels are particularly elevated in SM patients with a more severe disease phenotype characterized by organomegaly, BM hypercellularity, and increased serum tryptase [74,75]. Therefore, IL-6, although not specific for MC, can be considered a good diagnostic and prognostic biomarker of the severity of mastocytosis.

##### IL-31

MC can secrete IL-31, a well-known pruritogenic mediator, and circulating levels can be increased in mastocytosis and correlate with disease severity, especially intensity of pruritus, tryptase levels, osteosclerosis, and extent of BM infiltration by MC [76]. IL-31 levels can be influenced by sex and body mass index and can be elevated also in other diseases, such as atopic dermatitis, cutaneous lymphomas, and psoriasis. Further studies are needed to validate the use of IL-31 levels as a diagnostic marker in patients with mastocytosis, particularly in those with osteosclerosis and advanced or smoldering SM.

## 4. Flow Cytometry Biomarkers of MC Activation

Flow cytometry is a very useful tool to differentiate normal and neoplastic MC phenotypes, whereas its role in the assessment of the MC activation state is still under debate. However, the evaluation of surface molecules involved in MC activation may be potentially useful to investigate the MC activation state in MC disorders.

Flow cytometry is a widely diffused technique; it is standardized and allows the simultaneous visualization of multiple MC-associated markers. Other tools can be employed for the diagnostic workup of MC-related disorders, such as histopathology, morphology, and immunohistochemistry: these methodologies require tissue (e.g., skin or BM) biopsy, but allow the direct observation of MC infiltrates and their morphology.

MC identification by flow cytometry is challenging and requires high-standardized gating strategies for several reasons. First, MCs are present at very low frequencies in normal BM (0.008–0.0082% of all nucleated cells); therefore, a high number of events should be acquired for reliable and complete characterization. Second, MCs are heterogeneous in size and granularity with high autofluorescence, thus making their identification based on linear parameters (forward and side scatters) difficult [7]. Immunophenotype findings of normal MC can vary according to the type of tissues studied. CD117 is the fingerprint of normal MC: BM MC are identifiable by a constantly strong CD117 expression; however, the evaluation of other markers is useful for the correct identification and characterization of MC by flow cytometry [7,77,78].

Flow cytometry can be performed on heparinized or EDTA (ethylenediamine tetraacetic acid) BM blood specimens or fresh biopsies. For BM samples, a quick and vigorous BM aspiration is recommended to detach MC from BM stroma. For tissues, fragments should be finely cut (1–2 mm^3^), and mechanical disaggregation should be preferred to enzymatic procedures [77]. Samples should be processed and acquired within 3 h after collection or a maximum of 24 h if kept at +4 °C. Data acquisition should be performed in two steps: (1) acquire and record a minimum of 2–5 × 10^4^ events using the whole sample cellularity as recording gate; (2) append a minimum of 10^2^–10^3^ events using CD117+ live cells as recording gate [77,78]. CD45, CD117, and CD34 expression should be first investigated for MC identification, and an unstained sample should be included to assess basal autofluorescence. First, the CD117^+^ fraction can be enriched by gating on whole BM cellularity (side scatter vs. CD117), and subsequently, MCs can be identified based on CD117 and CD45 expression., as MCs are usually CD117^high^CD45^low^ cells [79,80,81]. BM myeloid progenitor population also shows CD117 positivity; however, these cells can be differentiated from MCs by further investigation of CD34 expression, as the progenitors are CD34^+^CD117^+^ while MCs are CD34^−^CD117^+^.

BM MCs are constantly positive for FcεRI, the high-affinity IgE receptor. The binding of the IgE receptor with IgE is one of the mechanisms involved in MC activation. FcεRI are expressed both on normal and clonal MCs, without differences between healthy controls and patients with indolent forms of mastocytosis [82]. Once identified, MCs should be characterized for normal or clonal immunophenotypes, as described below [80,81]. Figure 2 displays the gating strategy for MC identification and immunophenotyping.

### 4.1. Identification of Clonal MC by Flow Cytometry

D816V or other mutations in the *KIT* gene are considered the main driver of SM pathogenesis and are probably responsible for several immunophenotypic alterations; however, high immunophenotypic heterogeneity in SM may be attributable to additional mutations involving genes different from *KIT*. MCs obtained from BM of patients with SM show an increased SSC and greater level of autofluorescence compared to normal counterparts, with an expression of high internal complexity in terms of intracellular granules and high size [83]. In SM and monoclonal MCAS, MCs express aberrant surface markers, such as CD2, CD25, and CD35, or show modifications in the expression levels of molecules physiologically present on MCs, including CD63 or CD203c [81]. Although not always expressed on clonal MCs, CD2 and CD25 are constantly absent in normal BM MCs. CD35 reactivity cannot be pathognomonic of clonal MCs derived from ISM patients, because CD35 can be positive in BM MCs from B-cell malignancy [83]. The clinical relevance of these findings is demonstrated by the aberrant expression of CD2 and/or CD25 on MC, which is one of the minor WHO criteria for SM diagnosis [12].

CD25, the α chain IL-2 receptor, is typically expressed on CD4^+^FoxP3^+^ T cells with regulatory functions or on CD8^+^CD25^+^ T cells with a memory phenotype. Normal MCs do not express CD25, while neoplastic cells have increased levels of this receptor, especially in indolent SM [84]. Conversely, CD25 expression in well-differentiated systemic mastocytosis (WDSM), an SM variant associated with low MC burden and frequently lacking D816V mutation, is low or absent [85].

CD2, a transmembrane glycoprotein of the immunoglobulin superfamily, is found on T and natural killer (NK) cells, thymocytes, and dendritic cells. Normal MCs do not express CD2; conversely, neoplastic clones in indolent SM are positive for CD2 in about 80% of cases, while it can be absent or low expressed in well-differentiated forms [84]. In aggressive disease, MCs could be negative for CD2 while maintaining bright expression of CD25. Therefore, several studies propose CD25 as a more sensitive and specific immunophenotypic biomarker of all forms of SM compared to CD2, with a specificity of 99.2% and a sensitivity of 100% [86].

As discussed above, BM MCs from patients with indolent SM are also positive for CD35, although this marker is not pathognomonic of clonal MC [83].

CD30 is another aberrant marker expressed on neoplastic MCs, especially in WDSM [87]. CD30, a member of the tumor necrosis factor (TNF) receptor (TNFR) superfamily, is physiologically expressed by CD8^+^ T cells. Aberrant CD30 expression is found in several hematological and solid tumors [88]. CD30 expression is strongly associated with SM: more than 80% of patients with smoldering/aggressive SM or MC leukemia frequently show high CD30 surface expression, and also patients with ISM and tryptase levels ≥50 ng/mL have frequently positive and highly positive CD30 expression [87]. CD30 expression in neoplastic MCs could be related to the presence of an aberrant karyotype or *KIT* D816V mutation [89]. Moreover, high CD30 expression is found in patients with WDSM. For these reasons, CD30 has been recently proposed as a new additional minor criterion for SM diagnosis [12]. CD30 could be present in both indolent and advanced forms of SM, therefore it cannot be employed for differential diagnosis between indolent and advanced diseases, and CD30 is not considered a negative prognostic marker of SM [90].

BM MCs from patients with aggressive SM show high expression of HLA-DR and CD123 and decreased expression of CD117 and FcεRI and cytoplasmic tryptase [84]. Expression levels of CD117 and FcεRI could decrease with disease progression and aggressiveness, as neoplastic MCs progressively lose lineage-specific differentiation markers and acquire stemness characteristics [81]. Moreover, CD59 is frequently overexpressed in MC leukemia. Soluble CD117 and CD25 can be increased in the plasma of patients with either indolent or aggressive SM, especially with extensive marrow involvement and high total tryptase levels [91,92].

Other markers that can be found on neoplastic MCs are CD123, an α subunit of interleukin-3 receptors; adhesion molecules, such as CD11b, CD11c, CD18, CD22, CD49e, CD51, and CD54; complement-related CD55 and CD88; FcγRIIIB (CD16) and FcγRI (CD64) receptors [81]; and CD203c over-expressed on neoplastic MCs from SM patients. CD123 expression is increased on BM MCs from SM patients with poor prognosis and is associated with more immature phenotypes [84].

Table 4 summarizes the main surface MC markers used for diagnostic purposes on precursors of MC, normal MC, and different variants of mastocytosis.

### 4.2. Receptors Involved in MC Activation

The immune functions of MC are modulated by a variety of activating and inhibitory surface receptors. MCs are particularly rich in surface receptors that can induce the release of pre-stored mediators or synthesis of a number of lipid products or cytokines [5]. The main activating receptors on the MC surface include FcεRI, which activates intracellular signals responsible for the development of hypersensitivity reactions, purino-receptors (e.g., CD203c), adhesion molecules (e.g., CD29, CD33, and CD44), and tetraspanins, such as CD9 and CD63. CD203c is upregulated in response to IgE-receptor cross-linking and, as mentioned above, is also over-expressed on neoplastic MC of patients with SM.

CD63, a member of the lysosome-associated membrane glycoprotein (LAMP) family, is involved in vesicle fusion and is required for efficient degranulation upon IgE-mediated FcεRI activation [93]. CD63 is weakly expressed on resting MCs. After activation, CD63 aggregates with FcεRI and is rapidly upregulated together with CD203c, reaching a peak within 20–30 min after the allergen challenge [94,95,96]. The MC activation test, a promising tool in the diagnostics of IgE-mediated reactions, is based on the identification of activated MCs by up-regulation of CD63, CD107a, or CD203c [97]. Tetraspanins can also aggregate with other proteins forming the so-called tetraspanin-enriched microdomains composed of membrane receptors, kinases, integrins (CD29 or β1, CD49d or α4, α3, α6, CD18 or β2, and CD11a or αL forming the lymphocyte function-associated antigen 1 [LFA-1] in association with CD18), and other tetraspanins, such as CD9, CD81, CD82, and CD151. Normal MCs can express at low levels other adhesion molecules, such as CD11b (integrin αM), CD11c (integrin αX), CD22 (Siglec-22), CD49e (integrin α5), CD51 (integrin αV), and CD54 (intercellular adhesion molecule 1, ICAM-1), or complement molecules including CD55 (complement decay-accelerating factor), CD59 (membrane attack complex inhibition factor) and CD88 (C5a receptor), or immunoglobulin receptors, such as CD16 (FcγRIIIB), CD32 (FcγRII), and CD64 (FcγRI) [78,81].

Mas-related GPCR-X2 (MRGPRX2) is a newly discovered receptor associated with MC activation and is a member of the G protein-coupled receptor (GPCR) family [98]. MRGPRX2 is a multi-ligand receptor responding to various stimuli and induces MC degranulation in an alternative non-IgE-mediated mechanism through at least three signaling pathways: MEK1/2 and ERK1/2 MAP kinases via IKK; phosphatidylinositol 3 kinase (PI3K); and protein kinase C (PKC) via PLC and transient Ca^2+^ mobilization with single granule release [99,100]. MRGPRX2 is principally expressed on skin-resident MCs [101,102]. Different endogeneous and exogeneous substances can activate cutaneous MCs via MRGPRX2, such as neuropeptides, proteins derived from granules of eosinophils, antimicrobial peptides, opioids, and others [103]. MRGPRX2 can be involved in neurogenic inflammation, pain, and itch. Indeed, after stimulation of sensory nerve endings by MC-released tryptase, this receptor can be activated by substance P in a positive feedback loop, and triggered MCs can produce a large amount of IL-31, a well-known itch mediator [104]. Several studies attribute to MRGPRX2 a pathogenic role in the development of inflammatory skin diseases, such as chronic spontaneous urticaria (CSU) and atopic dermatitis (AD). Similar to other GPCRs, such as complement receptors C5AR1 and C3AR1, MRGPRX2 has a role in pseudo-allergic and anaphylactic reactions [105,106]. Several drugs, e.g., neuromuscular blocking agents (such as mivacurium) or antibiotics (such as fluoroquinolones) can induce reactions through the activation of MC degranulation via the MRGPRX2 pathway [106]. Giavina-Bianchi et al. reported a case of a patient with SM who developed an anaphylactic reaction in response to ciprofloxacin with negative results of specific IgE and skin testing [107]. Despite its biological interest in pseudo-allergic drug reactions and anaphylaxis, as well as its potential as a therapeutic target, MRGPRX2 is still not routinely employed in clinical practice as an MC activation marker; however, as a surface marker, MRGPRX2 could be quickly and specifically investigated on each MC subset by flow cytometry together with other activation and/or inhibitory molecules.

Inhibitory receptors on the MC surface include three members of the CD300 family: CD300a, CD300c, and CD300f [108]. MC, eosinophils, and basophils display CD300a on their surface, which can be activated by phosphatidylserine and phosphatidylethanolamine [109,110,111,112]. CD300a engagement causes decreased FcεRI- and DNAM-1-mediated MC activation by interfering with phosphatases and tyrosine kinase signaling pathways; in addition, CD300a reduces IgE- and stem cell factor-induced cell survival [113,114,115]. Similarly, CD300c reduces cytokine production and MC activation through phosphatidylethanolamine interaction [116,117].

### 4.3. Potential Markers of MC Activation

Despite MCs expressing multiple activating receptors on their surface, none of these receptors are validated for extensive use in clinical practice as a marker of MC activation. The research of potential biomarkers to assess the MC activation state represents a novel field to explore in future studies. Potential biomarkers of activation include CD63, CD69, CD203c, FcεRI, CD107a/b, and other molecules involved in MC degranulation [42].

CD63 is important for membrane fusion processes after FcεRI activation [93]. CD63 is abundant on vesicles from antigen-stimulated MC and seems the only tetraspanin involved in antigen-driven MC activation [118,119]. Up-regulation of CD63 after immune stimulation has been recently studied on MC progenitors (pMC), a cell population characterized by CD34^high^, CD117^+^, and FceRI^+^, extremely rare in peripheral blood [120]. When these cells are stimulated with anti-IgE, CD63 and LAMP-1 are upregulated. Two members of the LAMP family, CD107a/CD107b, also known as LAMP-1/2, are located in granules and are expressed at very low levels in resting MCs. LAMP-1/2 is quickly translocated on the plasma membrane after IgE stimulation, and its levels simultaneously increase with histamine, leukotriene C4, and prostaglandin D2 [94].

## 5. Conclusions

The activation of MCs is a key feature in a variety of allergic and inflammatory diseases. Primary MC disorders, such as mastocytosis and monoclonal MCAS, are associated with clinical signs and symptoms due to dysregulated mediator release. While classical triggers of MC activation, including Hymenoptera venom, food allergens, drugs, alcohol, and exercise, may have a role in MCAS, the main pathogenetic feature in these patients is probably an “intrinsic” MC hyperreactivity. Therefore, soluble and cell-associated biomarkers will be extremely useful to diagnose and manage MC disorders.

Circulating markers of MC activation and disease severity are histamine, tryptase, and other newly formed molecules, such as PGD2 or LTE4. Tryptase is the most specific marker of MC activation and of mastocytosis and is also used for risk stratification of advanced SM; while measurement of histamine metabolite levels in 24 h urine samples is an additional tool for monitoring global histamine turnover in SM and MCAS, especially for differential diagnosis of patients without skin lesions. The diagnostic and prognostic utility of other circulating biomarkers is still under investigation and larger studies are needed to validate the use of molecules, such as PGD2 or LTE4, for the diagnosis and monitoring of SM and related disorders.

Flow cytometry is routinely employed for the identification and characterization of normal and neoplastic MCs in health and diseases because surface marker expression can be studied at the single-cell level in various biological samples, including BM and skin biopsies. Indeed, neoplastic cells differ from their normal counterparts because of the expression of aberrant and/or stemness markers even in very low frequency (<0.001%) populations, making flow cytometry an important tool for diagnosis, disease classification, and monitoring of minimal residual disease. While surface molecules are very useful for diagnosis and prognostic stratification of mastocytosis, none of them has been demonstrated so far to be a good and reliable biomarker of MC activation. Unfortunately, the absence of mature MCs in the blood precludes the possibility of investigating these cells without performing invasive (tissue biopsy) procedures and, thus, hampers the possibility of studying surface markers in patients with MCAS. Despite these limitations, research to identify new MC-specific mediators or surface markers will continue to uncover reliable biomarkers of MC activation and to improve diagnosis and prophylactic/therapeutic management of patients with mastocytosis and MCAS.

## Figures and Tables

**Figure 1 ijms-24-07071-f001:**
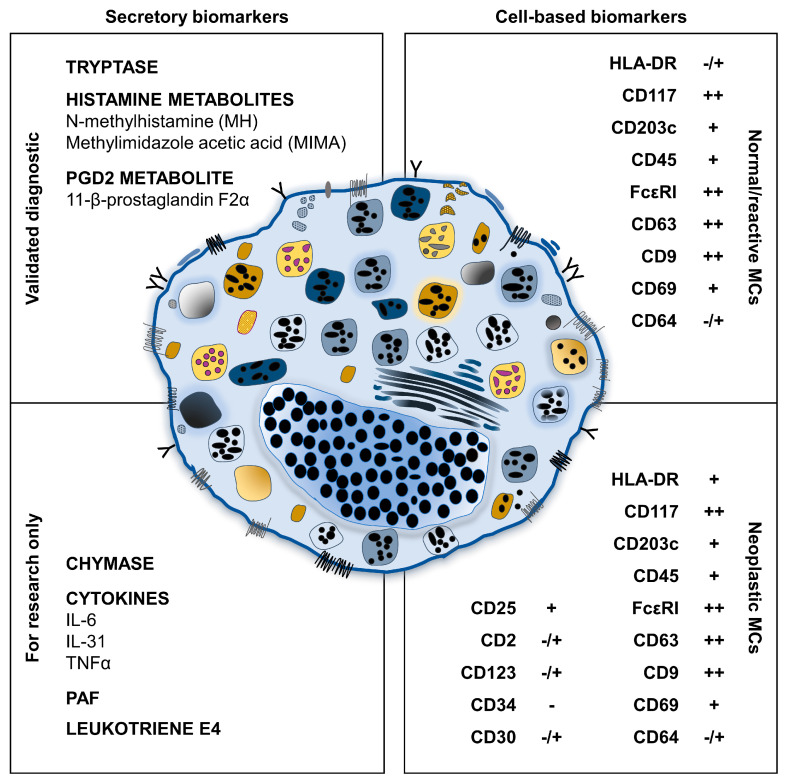
Secretory and membrane-associated MC biomarkers. Circulating biomarkers of MC activation are reported for molecules validated in clinical practice (upper left) or for research use only (bottom left). Cellular biomarkers of normal/reactive (upper right) and neoplastic (bottom right) MC are displayed. Abbreviations. PGD2, prostaglandin D2; IL, interleukin; TNFα, tumor necrosis factor alpha; PAF, platelet activating factor. ++, bright expression; +, expressed; -/+, dim expression; -, not expressed.

**Figure 2 ijms-24-07071-f002:**
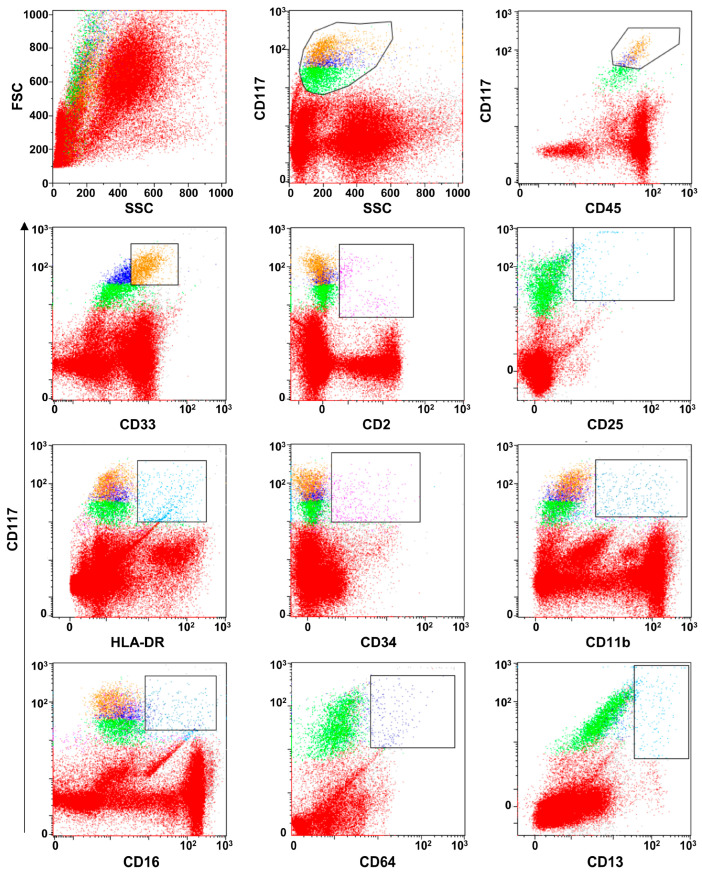
Flow cytometry gating strategy for MC immunophenotyping. MCs can be first identified as CD117^high^ cells with low side scattering (SSC) and CD45^high^ expression. On gated MCs, expression of normal markers, such as for CD33, CD16, CD11b, CD13, and CD64, of aberrant markers, including CD2 and CD25, or of increased expression of precursor makers, such as CD34 and HLA-DR, can be performed.

**Table 1 ijms-24-07071-t001:** Mast cell activation syndromes.

Disorder/Condition	Main Features	Clonality
Primary/monoclonal	*KIT* D816V mutation and/or CD25+ in most cases	Yes
Secondary	Caused by allergic or immunologic diseases	No
Idiopathic	Diagnostic MCAS criteria fulfilled with unknown causes	No

**Table 2 ijms-24-07071-t002:** Main features of mast cell (MC)-related disorders.

	Mastocytosis	MC Hyperplasia	Primary MCAS (Clonal)	Secondary MCAS	Hereditary α Tryptasemia
Increased number of bone marrow MCs	++	+/-	+/-	-	-
Enhanced release of MC mediators	+	+	++	++	+/-
*KIT* mutations or other genetic abnormalities	*KIT* mutations (e.g., D816V)	-	*KIT* D816V	-	*TPSAB1* duplication
CD2/CD25 expression on MC	+	-	+/-	-	-
MC morphology alterations	+ (spindle-shaped MC)	-	+/-	-	-
Increased basal serum tryptase	++	-	-	-	+
Increased acute serum tryptase	+/-	-	+	+	+

++, bright expression; +, expressed; +/-, dim expression; -, not expressed.

**Table 3 ijms-24-07071-t003:** Soluble mediators and their metabolites derived from MC.

Biomarker	Diagnostic Value	Potential Limitations
Tryptase (serum)	Specific for MC activation and proliferation Diagnostic and prognostic value in SM Acute (2–12 h after onset of symptoms) and basal measurement for MCAS diagnosis	Increased levels in: HAT End-stage renal failure Non-mast cell hematologic disease
Histamine metabolites (24 h urine collection)	Correlate with MC proliferation in SM Potentially useful for SM diagnosis in patients with slight elevation of serum tryptase and without skin lesions	Influenced by diet, bacterial contamination, and storage conditions Specific cut-off not established
PGD2 metabolites (24 h urine collection)	Correlate with symptoms in SM and MCAS Can guide decision to initiate aspirin therapy	Not recommended as a single test of MC activation Heterogeneity in proposed cut-off
Leukotriene E4 (24 h urine collection)	Correlate with symptoms in SM and MCAS Can guide decision to initiate antileukotriene therapy	Weak clinical evidence

**Table 4 ijms-24-07071-t004:** Normal and neoplastic MC immunophenotypes.

CD	BM Precursor	Normal MC	ISM	WDSM	ASM	Diagnostic Value
CD34	+	-	-	-	+	Correlates with disease aggressiveness
CD45	+	+				MC identification The expression could be reduced in aggressive forms
CD133	+					
HLA-DR	-	-/+	++	+	-/+	
CD117	+	++	+	+	-/+	
CD203c	-	+	++	+	-/+	Highly present in ISM/WDSM forms
FcεRI	-	++	++	++	-/+	
CD25	-	-	+	-/+	++	Aberrant on neoplastic MCs Highly expressed in aggressive forms
CD30		-	-/+	-/+	+	
CD123		-				
CD2		-	+	-/+	-	Aberrant on neoplastic MCs Mainly found in ISM
CD63		++	++	+	-/+	Activation markers Present on both normal MCs and ISM/WDSM
CD9		++				
CD18		+				Activation markers Candidate biomarkers of MCAS Not useful for differential diagnosis of SM
CD29		++				
CD33	+	++				
CD44		++				
CD49d		+				
CD49e		+				
CD51		+				
CD55		+				
CD59		+				
CD32		+				
CD69		+	++	+	+	

Abbreviations. CD, cluster of differentiation; MC, mast cell; BM, bone marrow; ISM, indolent systemic mastocytosis; WDSM, well-differentiated systemic mastocytosis; ASM, aggressive systemic mastocytosis. ++, bright expression; +, expressed; -/+, dim expression; -, not expressed.

## Data Availability

Not applicable.

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
