# Peer review of "Secretory and Membrane-Associated Biomarkers of Mast Cell Activation and Proliferation"

_ijms, 2023, doi:10.3390/ijms24087071_

Round 1

Reviewer 1 Report

Parente et al have summarized the mast cell markers that can be useful for identifying activation and disease state. The authors discuss extensively the strengths and limitations of clinical biomarkers for making diagnosis of various mast cell diseases. Overall the work is fairly complete and stays focused on the topic at hand.

Minor comments

1. Table 1 is somewhat hard to interpret. A more descriptive title (eg. features of MC-related disorders) would help. It is somewhat ambiguous what the row headers mean (eg. Bone Marrow MC -> does condition involve bone marrow MC or are bone marrow MC increased).

2. Line 121 - "estimated frequency of 4.4-6%" can the authors provide a source for this statistic?

3. line 216 -> "the raise of serum tryptase" should be "rise of serum tryptase" or "degree of increase in serum tryptase"

4. line 235 -> "can be easily monitoring in patients" to "can be easily monitored in patients"

5. Some of the tables are difficult to read without lines separating the rows. Possibly add lines or increase the margins between different rows to separate them better.

Reviewer 2 Report

This article by Parente et al. reviews various phenotypes of MCs and their mediators as clinical biomarkers for accurate diagnosis of MC-related disorders. MCs have been well-recognized as effector cells in allergic reactions, but other MC-related disorders, such as neoplastic SM and MCAS, have not been extensively discussed until recently. Because MC-related disorders may present with symptoms similar to other diseases, reliable diagnosis is critical for appropriate interventions. Thus, this article covers an important current topic.

The authors have collected valuable information in this review, first introducing MC-related disorders and their characterizations and describing MC mediators and surface molecules and their potential and utility as biomarkers. However, the article’s organization could be improved to deliver the information more effectively. A few specific suggestions are listed below.

1)    Table I contains information that will be described much later in the text. Referencing Table I in the Introduction section seems premature. Alternatively, Table I could summarize the clinical and pathophysiological characteristics of each disorder in a broader sense to familiarize the readers with the different types of MC-related disorders.

2)    In Section 2, “Definition of MC-related disorders,” the authors explain MC-related disorders as abnormal proliferation and activation characterized by MC burden, mutations, clonal markers, and tryptase levels. Although they continue the section by describing SM, MCAS, and HAT, it is difficult to discern which condition represents which category. Individual disorders could have a dedicated subsection (like the cytokine sections), providing featured pathophysiological and symptomatic characteristics with a simple and effective summary table like Table II for each.

In addition, discussing the molecular characteristics of each disorder before their detailed descriptions appear in the later sections seems ineffective. This section could be about a broad introduction to the different types of MC-related disorders with associated MC abnormalities and dysfunction and how they are similar to/different from allergic diseases. The inclusion of toxin-induced MC activation should also be considered for this section.

3)    The last paragraph in Section 2 about MC biomarkers may better serve as the beginning of the next section (lines 132-138).

4)    It may be a glitch during file conversion, but multiple question marks in rectangles were found on the surface of the mast cell in the middle of Figure 1. Also, the effectiveness of the mast cell diagram in the middle is not uncertain without the descriptions of the differently colored compartments.

5)    In Section 3, the authors describe the correlation of certain biomarkers with anaphylaxis. However, “anaphylaxis” is a rather broad term to describe severe systemic symptoms. Thus, specifying the symptoms (e.g., cardiovascular, respiratory, gastrointestinal, etc.) found to be associated with particular biomarkers would be helpful.

6)    Flow cytometry and its use for biomarker/phenotype detection are thoroughly described in a dedicated section. However, the reason why the authors decided to single out this technique to discuss as an immunophenotyping tool over other tests (e.g., cyto/histopathological analysis) is not apparent and should be explained.

7)    In Table 4, including the general functions of the surface markers an additional category for their diagnostic value as in Table III would be helpful. Also, please define the abbreviation for the SM types in the table legend.
